# Litter Management Strategies and Their Impact on the Environmental and Respiratory Microbiome Might Influence Health in Poultry

**DOI:** 10.3390/microorganisms10050878

**Published:** 2022-04-22

**Authors:** Dinka Ivulic, Ramon Rossello-Mora, Tomeu Viver, David A. Montero, Sonia Vidal, Francisco Aspee, Héctor Hidalgo, Roberto Vidal

**Affiliations:** 1Programa de Doctorado en Ciencias Silvoagropecuarias y Veterinarias, Campus Sur Universidad de Chile, Santa Rosa 11315, La Pintana, Santiago 8820808, Chile; ivulic.dinka@gmail.com; 2Marine Microbiology Group, Department of Animal and Microbial Diversity, IMEDEA (CSIC-UIB), 07190 Esporles, Illes Balears, Spain; ramon@imedea.uib-csic.es (R.R.-M.); tviver@imedea.uib-csic.es (T.V.); 3Programa de Microbiología y Micología, Instituto de Ciencias Biomédicas, Facultad de Medicina, Universidad de Chile, Santiago 8380453, Chile; davidmontero@med.uchile.cl; 4Programa de Inmunología, Instituto de Ciencias Biomédicas, Facultad de Medicina, Universidad de Chile, Santiago 8380453, Chile; 5Centro Integrativo de Biología y Química Aplicada (CIBQA), Universidad Bernardo O’Higgins, Santiago 8370993, Chile; 6Laboratory of Veterinary Vaccines, Department of Animal Biology, Faculty of Veterinary and Animal Science, Universidad de Chile, Santiago 8820808, Chile; svidal@vaccimed.cl; 7Agrícola Ariztía Ltda., Melipilla 9580752, Chile; faspee@ariztia.com; 8Laboratory of Avian Pathology, Faculty of Veterinary and Animal Sciences, Universidad de Chile, Santiago 8820808, Chile; 9ANID—Millennium Science Initiative Program—Millennium Nucleus in the Biology of Intestinal Microbiota, Santiago 8320000, Chile; 10Instituto Milenio de Inmunología e Inmunoterapia, Facultad de Medicina, Universidad de Chile, Santiago 8380453, Chile

**Keywords:** microbiome, broilers, tracheal, air, avian, litter, chickens, microbiota

## Abstract

Aerial and respiratory tract-associated bacterial diversity has been scarcely studied in broiler production systems. This study examined the relationship between the environmental air and birds’ respiratory microbiome, considering a longitudinal sampling. Total viable bacteria and coliforms in the air were quantified, and the 16S rRNA gene was sequenced from tracheal and air samples obtained through a novelty protocol. Air results showed a decrease in coliforms over time. However, at week 3, we reported an increase in coliforms (from 143 to 474 CFUc/m^3^) associated with litter management. Additionally, 16S rRNA gene results indicated a distinctive air microbial community, associated primarily with *Bacillota* phylum particularly of the *Bacilli* class (>58%), under all conditions. Tracheal results indicated a predominance of *Escherichia coli/Shigella* at the beginning of the productive cycle, shifting toward the middle and end of the cycle to *Gallibacterium*. However, at week 3, the dominance of *Escherichia coli/Shigella* (>99.5%) associated with litter aeration by tumbling stood out. Tracheal and air samples displayed a statistically different community structure, but shared differentially abundant features through time: *Enterococcus*, *Gallibacterium*, and *Romboutsia ilealis*. These results indicate the impact of production management protocols on the birds’ respiratory system that should be considered a breakpoint in poultry farm health.

## 1. Introduction

Poultry meat production is an essential source of nutrients worldwide [1]. Respiratory infections are common among these birds and are often detrimental, causing subclinical infections, mild respiratory symptoms, and economic losses [2,3,4]. It has come to be recognized that bacterial communities are important for maintaining respiratory health [5]. This respiratory microbiota can be conditioned by various factors that influence its composition, such as intrinsic factors of the host and extrinsic environmental factors [6] that can impact the microbiome and the subsequent progression of the disease.

Unlike what happens with the gastrointestinal microbiota, the microbiota of the respiratory system of broiler birds has been characterized by a few studies. Shabbir et al. [7] and Sohail et al. [8] were the first to use next-generation sequencing (NGS) to describe the lower respiratory tract microbiome in broilers, even though the scope of these studies was relatively small. Later, Johnson et al. [9] sought to define the baseline bacterial microbiota in the gastrointestinal, respiratory (tracheal), and barn environment (litter) of broiler birds. Here, the tracheal bacterial microbiota was comprehensively defined by comparing the effects of age, sample type, flock, and successive flock cycles. Another study was conducted using metagenomics to characterize the bacteria, eukaryotic viruses, bacteriophages, and fungi in a healthy broiler flock [10]. Age effects were seen for bacterial results, and Enterobacteria phage RB55 (Myoviridae family) correlated with *Gallibacterium* (*Enterobacterales* order) the last two weeks of fattening, showing the existence of bacteriophages that can influence bacterial diversity. This study provided a comprehensive view of the ecology of the avian respiratory microbiome and how an infectious agent can disrupt this environment, evolving into a disease. More recently, Zhou et al. [11] analyzed the effects of different levels of exposure to ammonia on the tracheal microbiota. The results showed that an increase in ammonia levels significantly decreases alpha diversity, changing the structure of the bacterial community, highlighting the effect of extrinsic environmental factors on respiratory microbiota.

The environment in intensive broiler production systems becomes relevant because it is characterized by a high animal density that facilitates the accumulation of feces and moisture in the litter. A floor rearing birds’ system creates favorable conditions for bacterial growth and inadequate ammonia concentrations that can lead to deciliation of the respiratory tract and increase susceptibility to infections [2]. Therefore, management is implemented by providing adequate shed ventilation and litter aeration by tumbling [12,13]. Taking this into account, we seek to study the tracheal microbiota and bacteria in ambient air of tunnel-ventilated broiler houses and an experimental animal facility at the beginning, middle, and end of the fattening cycle (42 days). The objective was to analyze the possible influence of ambient air bacteria on the respiratory microbiota of birds, considering litter aeration by tumbling in the experimental design. For this, the presence of viable bacteria in the air was quantified using the agar impact method (CFU/m^3^). In addition, the sequencing of the complete 16S rRNA gene was performed on tracheal and air (nitrocellulose membrane) samples, allowing maximum taxonomic resolution [14]. The study provides the first attempt to shed light on the influence of airborne bacteria on the respiratory microbiota of a broiler.

## 2. Materials and Methods

### 2.1. Experimental Design

Two tunnel-ventilated broiler houses (BHs) and one experimental animal facility (AF) were sampled three times during the fattening period, at weeks 1, 3, and 6. The BHs belong to the same farm, housing around 24,000 Cobb male broiler chickens within a vertically integrated broiler system. The floor was freshly covered with wood shavings at the beginning of bird fattening. Alternatively, control birds were reared from day 1 of age in the experimental animal facility unit for birds of the Avian Pathology Laboratory at the Universidad de Chile. These birds were obtained from the same hatchery as the birds destined for the BH system, subjected to the same immunization schedule (Appendix A), and fed from the same feed factory. Flocks were checked for clinical signs, and necropsy was performed to detect diseases or health issues.

Air samples were collected inside sheds using the agar impact method with automated equipment MAS-100 Eco^®^. Viable bacteria (CFU/m^3^) in the air were sampled using plate count agar (PCA) and McConkey plates in triplicate at each sampling time. For the NGS analysis, one nitrocellulose membrane per time was placed on a PCA plate to collect the sample. On the same day and in parallel with air sampling, birds were collected (*n =* 3) to perform a tracheal wash to study microbiota during growth (Appendix A).

The AF facility was used as a control, where four fans were located in each corner of the room (3.9 m in length, 3.4 m in width, and 2.5 m in height). Here, nine experimental birds plus three backup birds were housed in a wire cage at a density of below 10 kg/m^2^, where the birds were kept in a daily cleaning condition. For this, the sheets of blotting paper located at the cage’s base were changed daily. Feces adhering to the cage were removed and cleaned, and the room floor was kept free of dirt and moisture. In contrast, birds in the BH system were exposed to an environment where feces accumulated in the litter (wood shavings) throughout the six weeks of the production cycle. The two BHs were different at week three, where one was sampled 30 min after (BH-I) and the other before (BH-II) the management known as “litter aeration by tumbling”. This consists of breaking and turning the litter—using machinery—allowing it to aerate and dry the substrate [13,15]. This management was performed only once during the entire production cycle to reduce litter humidity during the rearing period (Figure 1).

### 2.2. Sample Collection

Healthy birds were euthanized by cervical dislocation after an overnight fast to avoid contamination of respiratory tissues by feed from the birds’ crops. The health status was corroborated by necropsy, and tracheal lavage was performed aseptically as previously described [16]. In brief, kelly-style hemostatic forceps were used to clamp the respiratory tract in two sites: below the larynx and above the carina of the trachea. Then, the tracheas were washed by pumping in and out of cold, sterile PBS five times. Volumes of 0.5 mL, 2 mL, and 5 mL were used at week 1, week 3, and week 6, respectively. The collected wash was centrifuged at 10,000× *g* for 10 min. The sedimented bacteria were stored at −80 °C until the subsequent DNA extraction.

In BHs—at the end of the shed and 40 cm from the ground—air samples were taken using the MAS-100 Eco^®^ impact aero-biocollector. In the AF, the measurement was taken in the center of the room at the height of the birds’ heads. For detecting viable microorganisms in the environment, 5 L of air was impacted on PCA, and 500 L of air was impacted in MacConkey agar plates. Moreover, we designed a novelty protocol: a 0.22 µm nitrocellulose membrane, 47 mm in diameter (Merck GSWG047S6), was aseptically placed on agar plate surfaces to impact a volume of 1000 L of air to recover the environmental DNA (on the air column). Then, they were then stored at −80 °C in 50 mL conical tubes for further processing.

### 2.3. Sample Processing

Total DNA was extracted from tracheal samples using the DNeasy Blood & Tissue Kit (Qiagen, MD, USA). The DNeasy Power Water Kit (Qiagen) was used for the nitrocellulose membrane DNA purification. Once the extracted DNA was purified following the manufacturer’s recommendation, quantity and quality (A260/A280) were measured by spectrophotometry (Epoch, BioTek, VT, USA). Additionally, the presence of the bacterial 16S rRNA gene amplicon was evaluated by DNA amplification using the primers 27F (5′-AGAGTTTGATCCTGGCTCAG-3′) and 1492R (5′-CGGTTACCTTGTTACGACTT-3′) followed by electrophoresis in agarose gel. Once the bacterial 16S rRNA gene had been proven, an extracted DNA aliquot was shipped to MR DNA (Molecular Research LP, Shallowater, TX, USA) for 16S rRNA gene library construction and sequencing. 

In the case of agar plates, incubation was carried out for 16 h at 37 °C, after which total colony-forming units (CFU) were counted. Feller’s statistical correction table was used according to MAS-100 Eco^®^ equipment procedures protocol to obtain the colony-forming units corrected (CFUc). Finally, the CFUc per cubic meter (1000 L) was calculated. 

### 2.4. Primers, 16S rRNA Gene Amplification, and Sequencing Process

Primers GM3 (5′-AGAGTTTGATCMTGGCTCAG-3′) and 23S1 (5′-GGGTTTCCCCATTCGGAAATC-3′) with a unique barcode linked to the forward primer were used in the generation of the sequenced amplicon. These primers encompass the entire 16S rRNA gene plus the adjacent internal transcribed spacer (ITS) located between 16S and 23S rRNA genes, which correspond to a region with a high degree of variability, allowing for additional studies on relatedness and discrimination of samples obtained from different sources [17]. A 35-cycle PCR (5 cycles for PCR products) was performed using the HotStarTaq Plus Master Mix Kit (Qiagen, MD, USA). PCR conditions were 94 °C for 3 min, then 35 cycles of 94 °C for 30 s, 53 °C for 40 s, and 72 °C for 90 s, with a final elongation step of 72 °C for 5 min. After, the success of amplification was checked in 2% agarose gel to determine the relative intensities of the PCR products. Next, multiple samples were pooled together in equal proportions based on their molecular weight and DNA concentrations. The PCR pool was purified employing Ampure PB beads (Pacific Biosciences, Menlo Park, CA, USA). The SMRTbell library preparation was achieved using the SMRTbell^®^ Express Template Prep Kit 2.0 following the manufacturer’s user guide. Finally, the sequencing procedure was completed at MR DNA (www.mrdnalab.com, Shallowater, TX, USA) on the PacBio Sequel system following the manufacturer’s guidelines.

### 2.5. Processing and OTU (Operational Taxonomic Units) Clustering

Upon completion of the initial DNA sequencing, each library underwent a secondary analysis, Circular Consensus Sequencing (CCS), using PacBio’s CCS algorithm (SMRT Link 9.0.0.92188). The CCS algorithm aligns the subreads individually from each template to produce consensus sequences, allowing the correction of stochastic errors caused in the initial analysis. Sequence data were then processed by applying the MR DNA analysis pipeline (MR DNA, Shallowater, TX, USA). In summary, barcodes were removed of the CCS sequencing data, then oriented 5′ to 3′, and sequences < 150 bp were discarded. Chimera removal and de novo OTU clustering were carried out using the usearch v.7 software package. Operational taxonomic units were defined by clustering at a similarity threshold of 97% (3% divergence). 

### 2.6. Phylogenetic Affiliation and OPU (Operational Phylogenetic Unit) Grouping

A total of 2172 representative OTU sequences were added to the nonredundant Silva 138 database using the ARB program package [18,19]. Sequences were aligned using SINA [20] and then inserted into a preexisting tree to select the closest relatives and finally to generate a phylogenetic tree reconstructed by Neighbor-Joining. The tree was finally modified using the parsimony tool to optimize the branching order. The parsimony tree was manually curated, generating 545 groups or OPUs. In this process, 396 OTUs noncodifying for rRNA genes were discarded. 

An OPU is the smallest monophyletic clade containing OTU representatives and their closest reference sequence. Where possible, OPUs included a type strain as a reference sequence. When the above was impossible, other available databases were used. For identity values > 98.7% that include a type strain sequence, amplicons were considered as belonging to the same species. When identity values were <98.7 and >94.5% with the closest relative type strain, amplicons were a different unclassified species of the same genus [21]. This approach uses phylogenetic inference instead of clustering by sequence identity to reflect diversity measures as group sequences in lineages that approach the species thresholds with a more robust view of microbial diversity [22]. 

### 2.7. Diversity, Comparative, and Statistical Analysis

Rarefaction curves and alpha diversity indices were calculated using the PAST software v4.05 [23] over prokaryotic reads of each sample. For data analysis, two samples were removed due to a low number of processed sequences: sample 2 on week 3 of BH-II and sample 2 on week 6 of the AF. Additionally, OPUs harboring < 10 sequences and those occurring in a single sample were discarded for beta diversity analysis and comparative analysis between each dataset. Data scaling was performed by the total sum scaling (TSS) method, transforming raw reads into relative abundances by dividing count data by the total number of reads in each sample. The MicrobiomeAnalyst server was used to perform the OPUs profiles analyses at different taxonomic levels and the β-diversity analyses [24]. Differential abundance analysis was carried out using Phyloseq v1.36.0 [25] and DESeq2 v1.32.0 [26] packages in R [27]. For air and tracheal datasets, DESeq2 “ratio” and “poscounts” size factor estimation were used, respectively. Venn diagrams were performed with the “InteractiVenn” web application [28].

Statistical differences between groups were determined using the stats v4.1.2 R package [27]. Normality was checked with the Shapiro–Wilk test and the homoscedasticity with the Fligner–Killeen test. Then, parametric (ANOVA) or nonparametric (Kruskal–Wallis) tests were performed. For post hoc pairwise comparisons, a Bonferroni or Tukey adjusted test was conducted.

## 3. Results

### 3.1. Viable Microorganisms in the Environment

Plate count agar (PCA) and MacConkey agar plates were analyzed to assess “total” viable bacteria and Gram-negative enteric bacteria on-air, respectively. In Figure 2A, the results of PCA are shown. According to the ANOVA test, the experimental animal facility (AF, control group) PCA count was significantly higher (*p* < 0.05) at week 3 and week 6 than week 1. By contrast, counts obtained from broiler houses (BHs) were out of the air sampler working range (over 400 colonies) due to the high bacterial load.

For MacConkey plate results (Figure 2B), the counts at week 1 in the BHs remained above the counts obtained in the AF. Next, an increase of more than three times in the counts at week 3 in BH-I stood out (from 143 to 474 CFUc/m^3^). The above coincides with the “litter aeration by tumbling” that was performed in BH-I 30 min before sampling. By contrast, BH-II at week 3, which did not have this litter management done, presented a lower count value. Finally, at the end of the cycle (week 6), all housing conditions remained similar and lower than 25 CFUc/m^3^ on average at the level of the AF.

These results showed a low presence of total bacteria in AF compared to the productive environment (BHs), which was impossible to measure on PCA due to the high bacterial load despite using the minimum sampling volume (5 L). A decrease in coliform counts on McConkey plates was observed in BHs throughout the study. The high count of cultivable coliforms in BH-I at week 3 is an important indicator of the high bacterial load we associated with the management of “litter aeration by tumbling” performed before sampling.

### 3.2. 16S rRNA Gene Analysis of Microorganisms in the Environment and Host

#### 3.2.1. Samples, Sequences, and OPU Distribution

Thirty-six samples were analyzed, corresponding to 9 environmental air nitrocellulose membrane samples and 27 tracheal wash samples. In addition, a process control for nitrocellulose membrane samples and a control for tracheal wash PBS were included, reporting zero 16S rRNA gene amplification for both. Of all the initial sequences generated by PacBio, those that did not correspond to the bacterial 16S rRNA gene were filtered. It was observed that the tracheal samples needed to eliminate a significantly larger number of sequences from the initial dataset than the air samples, which may be related to the presence of interfering eukaryotic DNA. Thus, the percentage of final filtered sequences regarding the initial sequences in the tracheal samples averaged 88.5% (±20.74%), corresponding to an average of 15,756 (±7861) sequences per sample (Appendix A). On the other hand, the percentage of final sequences with respect to the entire sequences in the air samples was higher than 99.9% in all samples, corresponding to an average of 15,379 (±8415) sequences per sample (Appendix A).

The reads obtained were grouped into OTUs at the 97% level of sequence identity, generating a total of 1771 different OTUs (856 tracheal OTUs and 1527 environmental OTUs) with an average of 70 (±68) OTUs for the tracheal samples and 568 (±165) OTUs for environmental samples (Appendix A, respectively). Phylogenetic inference using OTU representatives generated a total of 545 different OPUs (281 tracheal OPUs and 486 environmental OPUs) with a mean of 30 (±30) OPUs for tracheal samples and 195 (±58) OPUs for environmental samples (Appendix A, respectively). Of the 545 OPUs, 222 OPUs (40.7%) were shared between environmental and tracheal samples, 264 OPUs (48.4%) were present exclusively in environmental samples, and 59 OPUs (10.8%) were present only in tracheal samples (Figure 3A). After discarding OPUs harboring < 10 sequences and those occurring in a single sample, 107 OPUs (35.2%) were shared between environmental and tracheal samples, 160 OPUs (52.6%) were present exclusively in environmental samples, and 37 OPUs (12.2%) were present only in tracheal samples (Figure 3B).

#### 3.2.2. Tracheal Respiratory Samples Present Lower Diversity Than Environmental Air Samples

The rarefaction curves for the number of OTUs and OPUs and the number of reads analyzed are shown in Figure 3C. The OPU rarefaction curves approach showed a saturation point earlier than the OTU rarefaction curves, indicating a possible overestimation of diversity when using the traditional OTU approach [29]. Further, rarefaction curves of the tracheal samples saturated more rapidly than the environmental samples. Therefore, tracheal samples were dominated by fewer units—OTUs and OPUs—compared to the environmental samples across this study.

The environmental samples, showing a later saturation than the tracheal samples in the rarefaction curves, also showed greater alpha diversity in all times and groups (Figure 3D,E, Appendix A). Among tracheal samples, those at week 3 in BH-I and at week 3 in BH-II stood out, both showing the greater *Dominance* and lowest *Shannon*–*Weiner* index of diversity among all samples analyzed (Figure 3D,E). Notably, the tracheal samples from BH-I at week 3 showed a significantly higher dominance index close to 1 (0.997 ± 0.005), indicating that one taxon completely dominated the community (Figure 3E). In addition, the Shannon–Weiner index showed a significantly lower heterogeneity in these samples (0.012 ± 0.017) (Figure 3E). This behavior did not correspond to the diversity indices of the tracheal samples of AF at the same time, which showed a lower dominance and higher diversity, as with all other tracheal samples from week 1 and week 6 (Figure 3E, Appendix A).

The greater dominance and lower heterogeneity at week 3 of both the BH-I and BH-II tracheal samples occurred, although both sheds differed in the “litter aeration by tumbling” management. This dominance was characterized by tracheal OPUs, mainly of the phylum *Pseudomonadota*, but with a differential profile at the family level, as we see next.

#### 3.2.3. Environmental Air Shows a Distinctive Microbial Community

In order to analyze bacterial microbiota in air, we employed a heatmap of relative abundances, dendrogram, and principal coordinates analysis (PCoA) (Figure 4). The dendrogram and heatmap showed three main clusters with notable differences between samples of AF, BHs at week 1, and BHs at weeks 3 and 6 (Figure 4A). This result was further supported by the PCoA analysis, where the two main PCoA axes explained 88.1% of the variation (PERMANOVA, *p* = 0.002) (Figure 4B). A significant clustering pattern was observed on axis 1 (63.3%), where Bonferroni pairwise comparisons indicated three distinctive microbiome profiles between AF, BHs at week 1, and BHs at weeks 3 and 6. These results showed differences associated with housing type—AF vs. BHs—and time—BH-I at week 1 vs. BH at week 3 and 6—on air samples.

It should be noted that environmental samples at week 3 for both BH-I and BH-II were clustered together (Figure 4A), even though these two sheds differed due to the “litter aeration by tumbling” management. Thus, an increase in aerosols due to litter management may not affect the relative abundances of the features in the samples. However, litter management may increase absolute abundance through increased aerosols. It may reflect the “compositional nature” of microbiome data [30]. 

In general, relative abundances showed a predominance of *Bacillota* (>59%), particularly of the class *Bacilli* (>58%), under all conditions (Figure 4A, Appendix A). The predominance of *Streptococcus alactolyticus* and *Escherichia coli/Shigella* (a *Gammaproteobacteria*) were consistently observed in AF. In BHs, a new species of the class *Bacilli*, a new species of the genus *Erysipelatoclostridium*, *Erysipelatoclostridium spiroforme*, and a new species of the genus *Enterococcus* predominated at week 1. Then, a new species of the genus *Staphylococcus* and a new species of the genus *Corynebacterium* dominated at week 3. At week 6, a new species of the genus *Staphylococcus*, a new species of the genus *Corynebacterium*, and a new species of the family *Lactobacillaceae* were mainly present in BHs. These results indicated a distinctive microbial community, primarily associated with the *Bacilli* class.

#### 3.2.4. Tracheal Microbiota Composition Is Associated with Litter Management and Evolves in Broiler Houses

To compare the microbiota between housing at each sampling time (Figure 5A–C) and through time on each housing sampled (Figure 5D–F), beta-diversity and dendrogram analyses were performed. Significant differences were observed between housing experiments only at week 3 (Figure 5B and Appendix A), where the litter aeration by tumbling was carried out before sampling. Here, the two principal PCoA axes explained 78.3% of the variation (PERMANOVA, *p* = 0.009). A significant clustering pattern was observed on axis 1 (45.6%), where the post hoc test indicated two distinct microbiome profiles between BH-I—where litter management was carried out—and BH-II plus AF. A significant clustering pattern was also observed on axis 2 (32.7%), where the post hoc test indicated three distinct microbiome profiles between BH-I, BH-II, and AF. These differences are mainly explained by a high relative abundance of *Escherichia coli/Shigella* (OPU 437) in all samples in BH-I (>99.5%) and high relative abundance of an unclassified species of the genus *Gallibacterium* (OPU 447) in samples in BH-II (>98.3%) at week 3 (Figure 6). No differences were observed between housing at week 1 and week 6 (Figure 5A,C and Appendix A). 

In relation to the high relative abundance of *E. coli* (OPU 437) at week 3 in BH-I, it can be presumed that we were seeing the presence of the same clones in the trachea and in the air. This is because OPU 437, which was composed of 22 OTUs obtained through the sequencing of the complete 16S RNA gene plus ITS, shared a similar relative abundance profile of OTUs both in air and the trachea (Figure 7A). Here, the ITS region allowed samples obtained from different sources to be related given its high degree of variation in both length and sequence [17] (Figure 7B). Thus, there was no dominant *Escherichia coli/Shigella* OTU in BH-I at week 3, sharing a similar heterogeneous OTU population between the trachea and air, where OTUs 6, 7, and 10 covered over 88% of the relative abundance in both types of samples. We could infer from these results that the high presence of *E. coli* in the trachea at week 3 was not due to a pathogenic invasion, where a dominant pathogenic OTU or clone could be expected. Instead, we found evidence that reinforces the idea that *E. coli* established in the trachea came from the air associated with litter aeration by tumbling.

Next, significant differences were observed in BH-I and BH-II associated with time (age) (Figure 5D,E and Appendix A). For BH-I, the two principal PCoA axes explained 79.1% of the variation (PERMANOVA, *p* = 0.043) (Figure 5D). A significant clustering pattern was observed only on axis 1 (54.1%), where the post hoc test indicated two distinct microbiome profiles between week 3—where litter management was performed—and week 6. For BH-II, the two principal PCoA axes explained 87.1% of the variation (PERMANOVA, *p* = 0.026) (Figure 5E). A significant clustering pattern was observed only on axis 1 (64.1%), where the post hoc test indicated two distinct microbiome profiles between week 1 and weeks 3 and 6. In BH-I and BH-II, a transition from *Escherichia coli/Shigella* to *Gallibacterium* was observed as dominant representatives between the beginning and end of the productive cycle (Figure 6). No differences were detected in AF (Figure 5F), where dominance was not observed, with a high heterogeneity between replicas (Figure 6). In any case, a relevant presence of *E. coli* was observed in the tracheas at week 1 in AF, similar to that shown by the tracheas in BHs at week 1, which may be related to the common environment in the hatchery.

Altogether, these results indicated a predominance of *Escherichia coli/Shigella* (*Enterobacterales* of the family *Enterobacteriaceae*) at the beginning of the analyzed productive cycle, with a change toward the middle and at the end of the productive cycle to *Gallibacterium* (*Enterobacterales* of the family *Pasteurellaceae*). In addition, it was observed that the dominance of *Escherichia coli/Shigella* at week 3 in BH-I was associated with “litter aeration by tumbling”, which mechanically aerates the litter substrate on which the birds live and increases aerosols in the environment.

#### 3.2.5. The Composition of Environmental Air and Tracheal Microbiota Are Different but Share Differentially Abundant Features over Time

We performed PCoA analysis to evaluate differences in the community composition between tracheal and environmental air samples. The PERMANOVA (*p* = 0.001) showed clusters significantly different for the two groups, either when all samples were analyzed all together (Figure 8A) or when samples were analyzed by week of sampling or housing (Appendix A). These results followed the trend of alpha diversity analyses, showing two communities of different diversity and structure.

To identify significant features, we used the univariate DESeq2 method to analyze week 1 and week 6 tracheal and environmental samples, to explore the existence of differentially abundant OPUs for each dataset over time. A total of 81 differentially abundant OPUs were found in the air samples when comparing week 1 versus week 6 (*p*-adjusted < 0.05). On the other hand, five differentially abundant OPUs were found in the tracheal samples when the same comparison was made (*p*-adjusted < 0.05). Of these, three significantly different OPUs were shared between the two sample types: *Enterococcus* sp. (OPU 036), *Romboutsia ilealis* (OPU 313)*,* and *Gallibacterium* sp. (OPU 447) (Figure 8B). Both *Gallibacterium* and *Romboutsia ilealis* increased in the trachea and air between week 1 and week 6, whereas *Enterococcus* decreased between these two sampling times (Figure 8C). These three OPUs changed in the same direction among sample types, showing an association between bacterial species in the trachea and air when these were analyzed individually. Greater differences were observed in log2FoldChange between *Enterococcus* sp. of environmental air (4.6) and trachea (20.5), where the latter decreased to a greater extent. Then, the log2FoldChange observed between *Gallibacterium* sp. of environmental air (−6.2) and trachea (−18.0) showed a greater increase in this species in the trachea. Finally, *Romboutsia ilealis* showed a higher log2FoldChange in environmental air (−9.4) than in the trachea (−8.3). Thus, the conservative DESeq2 algorithm showed significantly different OPUs shared between tracheal and ambient air samples, despite the different composition of these two communities.

## 4. Discussion

Few studies have investigated the respiratory microbiota composition in broilers, and only one sought to analyze the effect of extrinsic environmental factors on respiratory bacterial communities [11]. Thus, we sought to investigate the presence of bacterial communities associated with aerosols in the air column and how these may change or influence the contents of the respiratory tract of birds during the fattening process. Consequently, this longitudinal study analyzed the tracheal and environmental air bacterial communities to shed light on the relationship between the two. This was performed in broiler houses where a new litter was used, which is easy to define in terms of its composition because it does not contain the bacterial DNA load of previous cycles, which can interfere when NGS data are analyzed. Litter aeration by tumbling management was included in the sampling design at week 3, which we consider can impact environmental conditions, not having been investigated to date. In addition, this litter turning within the production chain can cause alterations in the air column, which could have consequences at the respiratory level of the birds, if during the movement of litter, there are pathogens lodged on the floor (product of the birds’ excrement) that move to the air column.

The PCA results show a high number of total viable bacteria present in environmental air of commercial broiler houses, finding plate counts out of the operating range of the air sampler machine. Additionally, the AF increased the total bacterial load from week 3, probably associated with the increase in bird size, feces production, movement of the birds, and the consequent increase in aerosols. Further, this work is novel in describing the association of the management of “litter aeration by tumbling” with an increased count of Gram-negative enteric bacteria in the air. Even more, we were able to describe a decreased enteric count over time, finding similar counts at the end of the productive cycle (week 6) in the broiler houses and the experimental animal facility. Through observation, we associate this lower count of coliforms in the environment with a greater compaction of litter and less formation of aerosols from this at week 6. This compaction is generally caused by the accumulation of moisture over time from bird excreta and drinker spillage, among other factors [31]. So far, no previous data have been found on the association between litter compaction, aerosol formation, and a load of enteric bacteria in the air.

Much partial environmental data lead us to the following assumptions about the increased aerosols and consequent increase in the absolute number of bacteria in the air due to “litter aeration by tumbling”: (1) the increase of more than 3 times in the MacConkey counts at week 3 in BH-I (Figure 2B); (2) the simple observation after the incubation of the PCA plates where a higher density of colonies in BH-I is observed at week 3 (Appendix A); and (3) the highest number of environmental prokaryotic sequences obtained from ambient air (nitrocellulose membrane) in BH-I at week 3 (Appendix A). Together with tracheal data, these results lead us to believe that the most significant number of aerosolized bacteria are captured in the trachea and generate a higher concentration. The largest number of tracheal prokaryotic sequences obtained from the trachea in BH-I at week 3 (Appendix A), in addition to the observation of the 16S rRNA gene amplicon band intensity evaluated by DNA electrophoresis, where the BH-I samples at week 3 stand out from the rest of the samples (Appendix A), might explain this bacterial increase. Additionally, the fact that a similar OTU relative abundance profile is shared for *E. coli* OPU 437, in both air and trachea, reinforces this idea (Figure 7).

These results support the observation that the dominance of *Escherichia coli/Shigella* in the trachea in BH-I at week 3 is related to a high quantity of air bacteria due to litter management. However, the bacterial composition in the air did not differ at week 3 between BH-I and BH-II despite litter management. This highlights the importance of understanding that microbiome data are compositional and cannot report absolute abundance. Thus, selection in the trachea of *Escherichia coli/Shigella* from ambient air, where there is a high bacterial load and diversity, is presumably due to the preferential ability to colonize this niche. The high presence and capacity for colonization of *E. coli* could be enhanced by the higher bacterial load and also a temporal increase in ammonia levels related to the “litter aeration by tumbling,” which can take 24 to 48 h to return to normal levels [32]. These two factors, high bacterial load due to aerosol formation and ammonia, may play a significant role in broiler respiratory tract infections, with particular emphasis when potentially pathogenic *E. coli* such as APEC (avian pathogenic *Escherichia coli*) is present in the litter environment and environmental air. The ammonia effect was highlighted by a recent study, which reported that an increase in ammonia levels decreases alpha diversity and shifts the structure of the microbial community through inflammatory injury of the respiratory tract [11]. 

Next, we saw that despite the high *E. coli* load at week 3, the tracheal microbiota in BH-I evolves at the end of the fattening cycle to a profile clearly defined by *Gallibacterium* sp., as in BH-II at week 6. Therefore, we can interpret that this high dominance of *E. coli* at week 3 is transient, and mucociliary and immune clearance of the tracheal epithelium [6] restore the previous condition before litter aeration by tumbling. It is noteworthy that although a time-changing microbial structure was found in BHs, no change was observed in AF over time (Figure 5F). These results may be related mainly to the daily cleaning carried out in this facility, avoiding contact with aerosols and ammonia by these birds. 

Previously, Johnson et al. [9] reported a tracheal microbiota distinct from those obtained from gut and litter samples. Here, we show a different microbial composition between the trachea and environmental air in all housing types and sampling times. In addition, we observed a low alpha diversity of bacteria in the trachea compared to ambient air. Although the diversity and general structure of the bacterial communities are different, both communities show some particular OPUs in common. We observed three differentially expressed OPUs corresponding to bacteria adapted to living in the animal host: *Enterococcus* sp., *Gallibacterium* sp., and *Romboutsia ilealis*.

*Enterococcus* sp. (OPU 036), a facultative anaerobe genus, decreased in the trachea and environmental air between week 1 and week 6. It can survive under strict environmental conditions in the gastrointestinal tract, making persistence easier [33]. Its decrease in environmental air may be related to the progressive compaction of the litter over time due to the accumulation of humidity, preventing the aerosolization of fecal bacteria found in the litter. The above is supported by a more significant decrease measured as a log2fold change in tracheal samples than environmental air, showing a gradient of dissemination.

*Gallibacterium* sp. (OPU 447) increased in the trachea and environmental air between week 1 and week 6. This genus is known to be a typical inhabitant of the respiratory tract [34] and facultative anaerobic [35]. Mulholland et al. [10] previously reported the *Gallibacterium* sp. high presence at the end of the fattening cycle. Its high relative abundance in tracheas at week 6 means it is most likely spread from the trachea into the ambient air. This observation is supported by a more significant increase at week 6 measured as a log2fold change in tracheal samples, compared to environmental air.

*Romboutsia ilealis* (OPU 313) is an obligately anaerobic bacteria commonly detected in the gastrointestinal tract [36]. This species increased more in the environmental air than in the trachea, albeit slightly. As *R. ilealis* is anaerobic, it might be expected that this bacterium would not colonize the trachea and would be present transiently.

Broiler birds are constantly exposed to litter aerosols and bacteria carried in them. We should expect exposure to the litter environment and preferential selection for microbes from this environment with the capacity to colonize. Additionally, some bacterial species may likely be present transiently. Whether there is an establishment or transitory passage of microorganisms through the respiratory tract of birds, they can have a significant effect on their health that requires further investigation. Thus, this work provides new knowledge regarding the influence of ambient air and the bacteria present in it on the respiratory microbiota of broiler chickens. Additionally, we found in litter management a critical point that can significantly influence the respiratory microbiota in broilers through the generation of a higher bacterial load in the air, not previously analyzed, and it must be considered a dissemination point of putative microbial pathogens.

## 5. Conclusions

Most studies of respiratory microbiomes in broilers have focused mainly on describing the host bacteria, but the influence of the ambient air microbiome on the respiratory system has not been investigated so far. This is particularly important under conditions where large numbers of animals are housed at high densities. Here, we describe the importance of aerosol formation and the management that can increase it, and its capacity to generate a change in the diversity of the tracheal microbiome. Considering this, we recommend that this point should be taken into consideration in diagnostic and prevention by isolation and characterization of bacteria in circumstances where they may cause disease. This is particularly valid for the case of infections with avian pathogenic *E. coli* (APEC), described as a pathogen that generates the greatest economic losses in the poultry industry [2]. These pathogenic *E. coli* could be differentiated from commensal *E. coli* according to specific virulence determinants, as described previously [37]. Thus, our results show that it is important to avoid misinterpreting the isolation of *E. coli* in abundance from the trachea as colonization and invasion, as we demonstrate that there may be a transient establishment in the trachea related to litter aeration by tumbling and aerosol formation. Hence, the knowledge of the influence of ambient air on the respiratory microbiome can help improve the diagnosis and prevention of diseases in broiler chickens.

## Figures and Tables

**Figure 1 microorganisms-10-00878-f001:**
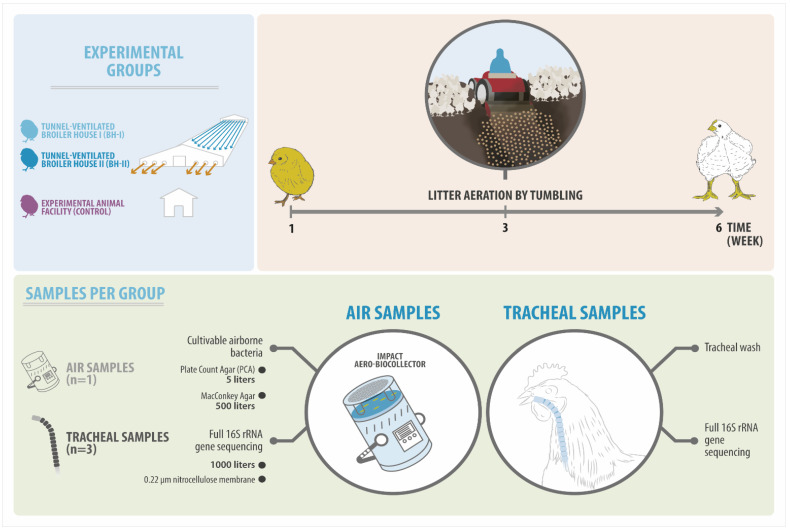
Experimental design of this study. Three experimental groups, including two sheds (BH-I and BH-II) and one experimental animal facility (AF), were sampled over time (week 1, 3, and 6). Air samples were taken to determine the load of cultivable bacteria and airborne bacterial diversity (16S rRNA gene). On the other hand, tracheal samples were taken to determine bacterial diversity (16S rRNA gene). It is important to highlight that in week 3, the sampling in the BH-I was performed 30 min after the litter aeration by tumbling.

**Figure 2 microorganisms-10-00878-f002:**
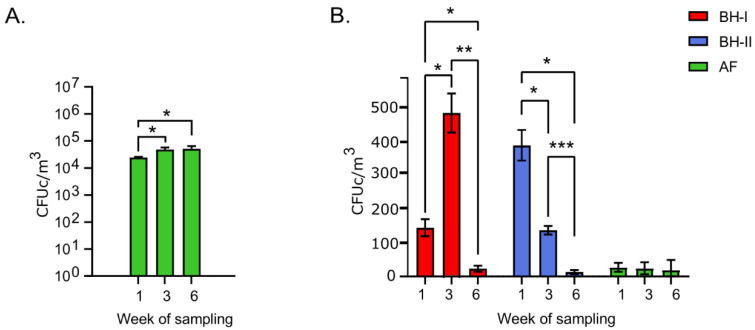
Colony counts of viable microorganisms. (**A**) PCA CFUc/m^3^ per week in experimental animal facility (*n =* 3). (**B**) MacConkey CFUc/m^3^ per week in tunnel-ventilated broiler houses and experimental animal facility (*n =* 3). Error bars depict standard deviations. Differences between week of sampling per facility were analyzed by ANOVA and Tukey’s post hoc. Asterisks (*) indicate significant differences with * *p* < 0.05, ** *p* < 0.005, *** *p* < 0.0005. AF: experimental animal facility, BH: broiler house.

**Figure 3 microorganisms-10-00878-f003:**
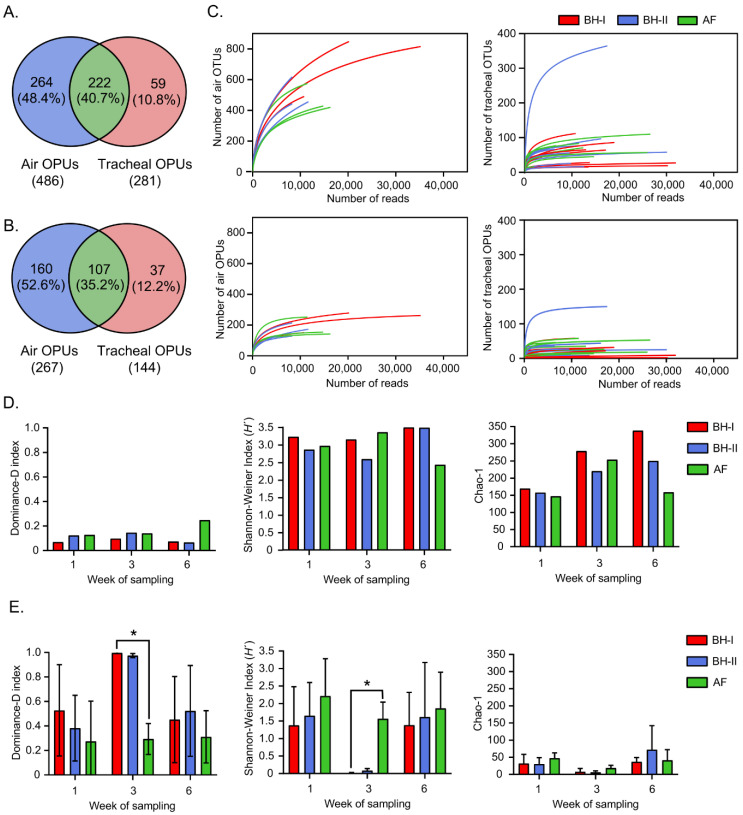
OPU distribution, rarefaction, and alpha diversity. Venn diagram before (**A**) and after (**B**) discarding OPUs for air and tracheal samples harboring <10 sequences and occurring in a single sample. (**C**) Rarefaction curves for detected OTUs and OPUs, where each line represents an independent sample. Alpha diversity indices for air samples (*n =* 1) (**D**) and tracheal samples (*n =* 3) (**E**), where the average of the tracheal diversity index ± standard deviation for each group is plotted. Differences (* *p* < 0.05) between facilities per week of sampling were analyzed by Kruskal–Wallis and Bonferroni’s post hoc. AF: experimental animal facility, BH: broiler house.

**Figure 4 microorganisms-10-00878-f004:**
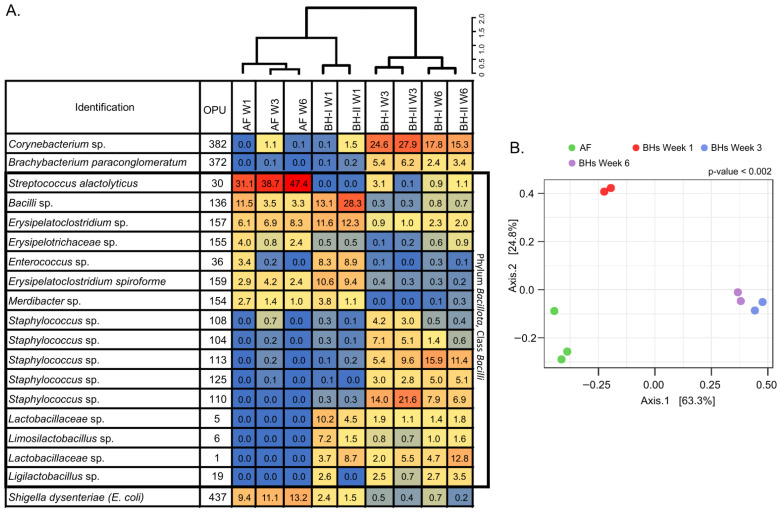
Heatmap, dendrogram, and beta-diversity analyses of air samples. Heatmap of relative abundances (%) of top OPUs per sample up to 5% (**A**). The inset color of blue indicates zero; yellow to increasing red values indicate above 1%. On top, dendrogram analysis, Bray–Curtis distance method, and Ward’s clustering algorithm. A list of OPUs and their closest relative sequence is given in Appendix A. (**B**) PCoA of samples using Bray–Curtis measures of beta diversity. Statistical: PERMANOVA. AF: experimental animal facility, BH: broiler house, W1: week 1, W3: week 3, W6: week 6.

**Figure 5 microorganisms-10-00878-f005:**
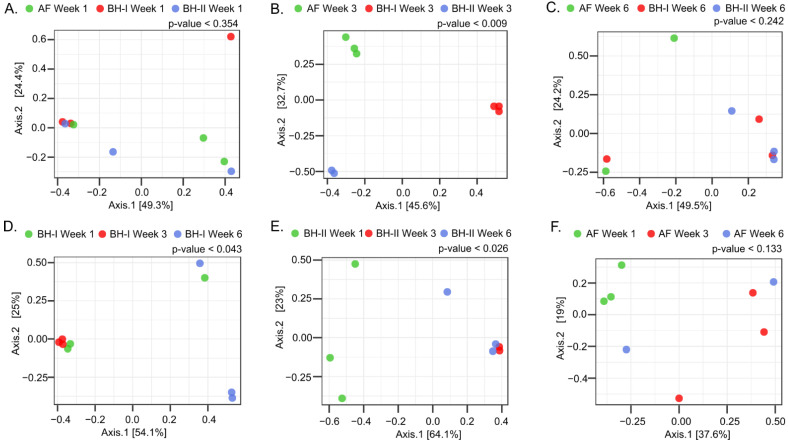
Beta-diversity analysis of tracheal samples. PCoA of samples by time and housing. In the top panel, analysis between housing at week 1 (**A**), week 3 (**B**), and week 6 (**C**). In the bottom panel, analysis over time of sampling on BH-I (**D**), BH-II (**E**), and AF (**F**). The analysis was performed using Bray–Curtis measures of beta diversity. Statistics: PERMANOVA. AF: experimental animal facility, BH: broiler house.

**Figure 6 microorganisms-10-00878-f006:**
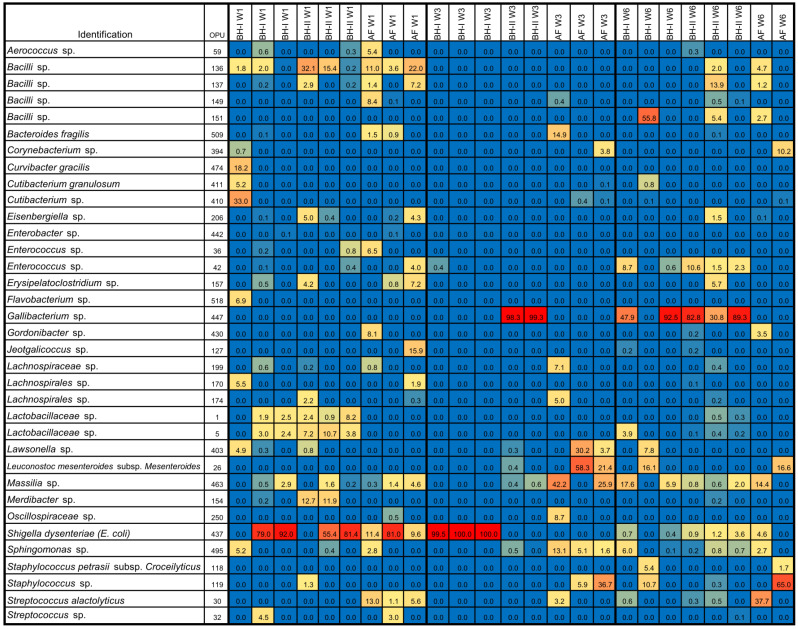
Heatmap of tracheal samples. Heatmap of relative abundances (%) of top OPUs per sample up to 5%. The inset color of blue indicates zero; yellow to increasing red values indicate above 1%. A list of OPUs and their closest relative sequence is given in Appendix A. AF: experimental animal facility, BH: broiler house. W1: week 1, W3: week 3, W6: week 6.

**Figure 7 microorganisms-10-00878-f007:**
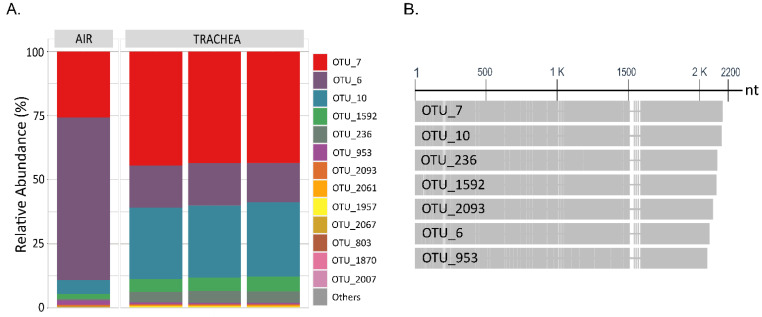
Composition of OPU 437 *Escherichia coli/Shigella* at the OTU level. (**A**) Stacked bar plot of relative abundances at the OTU level of OPU 437, where OTUs with counts <10 reads were merged into “Others” category. “Air” (*n =* 1) and “Trachea” (*n =* 3) depict samples in BH-I at week 3 after litter aeration by tumbling. (**B**) Graphical display for the alignments of nucleotide sequences of the 7 most abundant OTUs of OPU 437 in BH-I at week 3, where the ITS region is shown from the 1500 nucleotide (nt) onwards.

**Figure 8 microorganisms-10-00878-f008:**
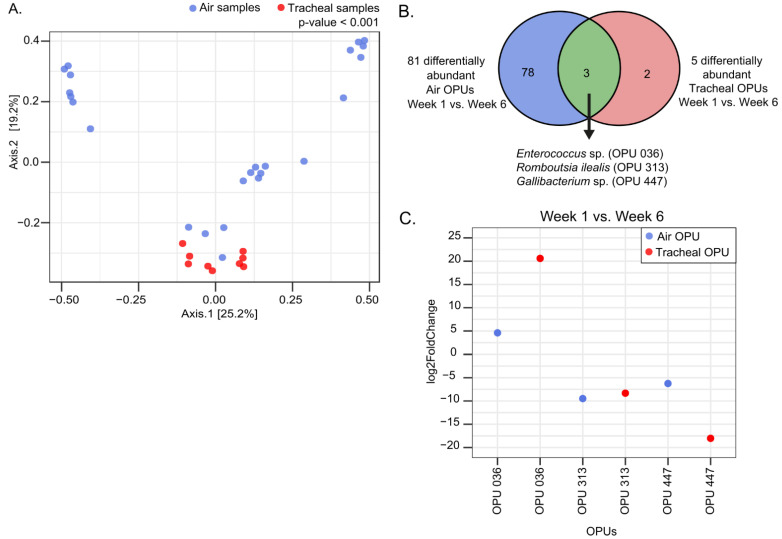
Beta-diversity and differential abundance analyses of environmental air and tracheal samples. (**A**) PCoA of environmental and tracheal samples using Bray–Curtis beta diversity measures. Statistics: PERMANOVA. (**B**) Venn diagram of OPUs analyzed in DESeq2 for air and tracheal samples. (**C**) DESeq2 analysis results indicating the Log2fold change of significant bacterial OPUs (*p*-adjusted < 0.05) in environmental air and tracheal samples.

## Data Availability

The data presented in this study are available on request from the corresponding author.

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
