# Peer review of "Litter Management Strategies and Their Impact on the Environmental and Respiratory Microbiome Might Influence Health in Poultry"

_microorganisms, 2022, doi:10.3390/microorganisms10050878_

Round 1
Reviewer 1 Report
Microorganisms – 1678894 “Litter Management Strategies and Their Impact on the Environmental and Respiratory Microbiome Might Influence Health in Poultry”
General Comments: The authors have use tracheal lavage to recover bacteria from broilers housed in commercial and research animal facilities.
The volume of PBS injected and recovered should be provided.
It is unlikely that the rinsate represents recovery from the air sacs and the lungs, so the sample is most likely only from the tracheal and the primary and secondary bronchi and does not represent the entire “Respiratory Microbiome”. Therefore, the title and text should use the term “tracheal microbiome” as in L564 of the Conclusions.
Additional information should be provided describing the housing and sampling in both BH and AF.
Specific comments:
L89 Delete “fattening”, since most broilers are brooded for the first 2 weeks (sample 1 was at 1 week) and the week 3 samples would be during the growing period, and only the final samples at week 6 during the finishing period could be considered the “fattening period”.
L92 Replace “fattening” with “brooding” or with “prior to placement of the chicks”.
This is where additional information regarding housing should be provided:
What at the time period between placing the shaving and the chicks into the BH?
What was the house configuration partial or whole house chick brooding?
If particle house brooding was used, when were the chicks provided access to the whole house.
What were the outside and inside house temperatures which would influence the house ventilation air flows during sampling?
For the AF house broilers, provide the dimensions of the house, the number of chicks placed, if the floor was covered with shavings, and the exhaust ventilation location in relation to the air sampling location.
L96, L121 Replace “food” with “feed”.
L97 Replace “discard” with “determine” or “detect”.
L104 Describe the “cleaning condition”.
This implies that these broilers were housed in wire cages, if so, describe caging facility.
L105-107 This sentence also implies that the AF broilers were housed in cages.
“In contrast, birds in the BH system were exposed to an environment where feces accumulated in the litter (wood shavings) throughout the six weeks of the production cycle.”
Provide a reference that describes “litter aeration” at week 3 is a common commercial practice.
Reference 13 was conducted in a research facility with 800 chicks place per room.
“The rearing was done inside an experimental poultry house in the Animal Research and Technology Centre (Instituto Valenciano de Investigaciones Agrarias, Segorbe, Spain) to mimic the real conditions of poultry production. The experimental house was tested for Salmonella before the experiment. In each rearing, 2,400 one-day-old chickens were received and divided equally in 3 experimental rooms.”
Also, for reference 13 the litter aeration was conducted starting at week 4 and then weekly “LA was carried out at weekly intervals (from wk 4 until wk 7) using a machine designed for this purpose (Benza, ER73AV, La Coruña, Spain, Figure 1).”
Explain why it was necessary in your BH sampling to aerate the litter at week 3.
L120-121 Provide a description of the housing of the birds for “overnight fast”
Was this done in the same BH and AF air space?
Were the birds provided access to water?
L121 Replace “respiratory” with “tracheal”.
L123 Provide the volumes of PBS injected and recovered for the tracheal lavage.
Reference 15 injected 10 mL and recovered 4mL.
L126 Here is where the house temperature and the air velocity or the number of exhaust fans operating in the BH need to be provided.
L128 Again the phrase “at the height of the birds head” implies cage units for AF.
L238 The results presented in Figure 2 B need to explain the high level of recovery at week 1 for BH-II, compared to BH-I at week 1, and to BH-II at week 3.
Author Response
We would like to thank Reviewer 1 for her/his advice and suggestions on the revision of our manuscript, and we hope that we have met their expectations.
The following are our specific responses to the reviewer's comments.
General Comments: The authors have use tracheal lavage to recover bacteria from broilers housed in commercial and research animal facilities.
The volume of PBS injected and recovered should be provided.
Response: The protocol implies minimal loss of PBS. A brief description of the protocol was included in the manuscript (L129–132).
“In brief, kelly-style hemostatic forceps were used to clamp the respiratory tract in two sites: below the larynx and above the carina of the trachea. Then the tracheas were washed by pumping in and out of cold, sterile PBS five times. A volume of 0.5 mL, 2 mL, and 5 mL was used at week 1, week 3, and week 6, respectively”.
It is unlikely that the rinsate represents recovery from the air sacs and the lungs, so the sample is most likely only from the tracheal and the primary and secondary bronchi and does not represent the entire “Respiratory Microbiome”. Therefore, the title and text should use the term “tracheal microbiome” as in L564 of the Conclusions.
Response: Although we understand the reviewer's comment, we do not agree with him. When we refer to respiratory microbiota, we keep in mind that the trachea is part of this system, which is evident throughout the manuscript. Other research articles have also followed this line of thought. It is understood that samples were taken circumscribed in anatomical zones within the respiratory system, i.e., Metagenomic Analysis of the Respiratory Microbiome of a Broiler Flock from Hatching to Processing. Microorganisms 2021, 9, 721. https://doi.org/10.3390/microorganisms9040721. Here the authors used "tracheal swabs" for sampling.
Additional information should be provided describing the housing and sampling in both BH and AF.
Response: The authors have addressed the above in other points, which also requested the same by the reviewer. A supplementary table summarizing the distribution of birds for each study group, BH and AF (Table S2), has also been incorporated.
Specific comments:
L89 Delete “fattening”, since most broilers are brooded for the first 2 weeks (sample 1 was at 1 week) and the week 3 samples would be during the growing period, and only the final samples at week 6 during the finishing period could be considered the “fattening period”.L92 Replace “fattening” with “brooding” or with “prior to placement of the chicks”.
Response: We do not agree with the reviewer's comment and would like to explain why. We use the term "fattening period" in its broadest context. This context is commonly used in scientific articles in journals such as "Poultry Science," known as the authoritative source for a wide range of poultry information, i.e., Liu C, Wang P, Dai Y, Liu Y, Song Y, Yu L, Feng C, Liu M, Xie Z, Shang Y, Sun S, Wang F. Longitudinal monitoring of multidrug resistance in Escherichia coli on broiler chicken fattening farms in Shandong, China. Poult Sci. 2021 Mar;100(3):100887. doi: 10.1016/j.psj.2020.11.064. Epub 2020 Dec 8. PMID: 33516478; PMCID: PMC7936140. Therein, the authors use the same concept to refer to "the fattening period of all chickens examined was approximately 40 d and these were sampled in 3 time periods (Laube et al., 2013); the first sample collection was between 1 and 2 d of age, the second between 17 and 20 d, and the third between 35 and 40 d."
This is where additional information regarding housing should be provided:
What at the time period between placing the shaving and the chicks into the BH?
Response: The elapsed time is 5 to 7 days before the arrival of the chicks.
What was the house configuration partial or whole house chick brooding?
Response: The house configuration is partial. The chick reception area accounts for 35% of the total area of the shed.
If particle house brooding was used, when were the chicks provided access to the whole house.
Response: After 20 days, the entire area was occupied.
What were the outside and inside house temperatures which would influence the house ventilation air flows during sampling?
Response: Outside, it varies from 10 to 35° C (Spring). Inside, the automated ventilation system is set to reach target temperatures based on age.
For the AF house broilers, provide the dimensions of the house, the number of chicks placed, if the floor was covered with shavings, and the exhaust ventilation location in relation to the air sampling location.
Response: We agree with the reviewer's comment. It has been addressed in lines 104 to 107. " The AF facility was used as a control, where four fans were located in each corner of the room (3.9 m in length, 3.4 m in width, and 2.5 m in height). Here, nine experimental birds plus three backup birds were housed in a wire cage at a density of below 10 kg/m2, where the birds were kept in a daily cleaning condition."
L96, L121 Replace “food” with “feed”.
Response: This has been done, Line 96 and 127
L97 Replace “discard” with “determine” or “detect”.
Response: This has been done, Line 97
L104 Describe the “cleaning condition”.
Response: It was addressed in lines 107 to 109. "For this, the sheets of blotting paper located at the cage's base were changed daily. Feces adhering to the cage were removed and cleaned, and the room floor was kept free of dirt and moisture."
This implies that these broilers were housed in wire cages, if so, describe caging facility.
Response: It was explained in lines 104 to 107.
L105-107 This sentence also implies that the AF broilers were housed in cages.
Response: It has been explained in Lines 105 to 107. “In contrast, birds in the BH system were exposed to an environment where feces accumulated in the litter (wood shavings) throughout the six weeks of the production cycle.”
Provide a reference that describes “litter aeration” at week 3 is a common commercial practice.
Response: We have added a new citation. Reference 15 is added in Line 116, which together to reference 13, both explain the Litter aeration process.
Reference 13 was conducted in a research facility with 800 chicks place per room.
“The rearing was done inside an experimental poultry house in the Animal Research and Technology Centre (Instituto Valenciano de Investigaciones Agrarias, Segorbe, Spain) to mimic the real conditions of poultry production. The experimental house was tested for Salmonella before the experiment. In each rearing, 2,400 one-day-old chickens were received and divided equally in 3 experimental rooms.”
Response: This reference also describes what the litter aeration process or technique consists of.
Also, for reference 13 the litter aeration was conducted starting at week 4 and then weekly “LA was carried out at weekly intervals (from wk 4 until wk 7) using a machine designed for this purpose (Benza, ER73AV, La Coruña, Spain, Figure 1).”
Response: This reference also describes what the litter aeration process or technique consists of.
Explain why it was necessary in your BH sampling to aerate the litter at week 3.
Response: It is the established protocol on the farm when using the new litter. When the litter has had more cycles of use (used litter), 2 to 3 turns can be used, depending on the "caking" of the litter. Caking is avoided because it generates problems in birds, such as "footpad dermatitis" (Reference 31 of the manuscript).
L120-121 Provide a description of the housing of the birds for “overnight fast”
Response: For the overnight fast, birds were placed on the floor over sheets of absorbent paper in a cleanroom of the Avian Pathology Laboratory of the University of Chile, enabled for this purpose.
Was this done in the same BH and AF air space?
Response: No
Were the birds provided access to water?
Response: Yes, but the water was removed the next morning, 2 hours before the trachea sampling.
L121 Replace “respiratory” with “tracheal”.
Response: Previously, it has been explained why not by the authors.
L123 Provide the volumes of PBS injected and recovered for the tracheal lavage.
Response: It was clarified in Lines 131 to 132.
Reference 15 injected 10 mL and recovered 4mL.
Response: It was clarified in Lines 129 to 132.
L126 Here is where the house temperature and the air velocity or the number of exhaust fans operating in the BH need to be provided.
Response: The automated ventilation system is configured to reach the desired temperatures based on age. Air velocity and the number of exhaust fans operating were automated parameters that were not recorded, but samples were taken at the same time in spring.
L128 Again the phrase “at the height of the birds head” implies cage units for AF.
Response: It was clarified in Lines 105 to 107.
L238 The results presented in Figure 2 B need to explain the high level of recovery at week 1 for BH-II, compared to BH-I at week 1, and to BH-II at week 3.
Response: The level of recovery at week 1 for BH-II was lower than for BH-I at week 1. It may be due to a variation between samples at that time since both samples were taken under the same conditions. The sample level at week 3 should be compared with samples taken simultaneously to avoid confounding variables.
Reviewer 2 Report
Dear Authors
I really appreciated your study for several factors. As first the originality : in general, the microflora of the respiratory tract of chicken is poorly investigated in spite of the numerous diseases affecting the chicken trachea, often with "collaboration" of a poor quality of litter . In this context, studies on turkey are more numerous . An other important factor is represented by the interrelationships among the gut , air and respiratory microflora influencing air and litter quality (also conditioned by possible treatments, as tumbling) . I would define it as a real circuit. Your results are also supported by in-depth statistical analyses and discussed in exhaustive manner . Moreover data relating to Escherichia coli and Gallibacterium spp.( Gallibacterium anatis is currently considered an emerging pathogen) are interesting , also because of the potential role played as carriers/spreaders of resistance genes. Finally, although your results need to be further supported by other investigations , it seems clear that the environment air as well as the litter can influence microbiome of tracheal and intestinal tract respectively , although I believe that everything is strictly associated . Following you can find minor changes I recommend you
Material and methods section It should be correct to specify vaccination plains. They could be an influencing factors for air and respiratory microbiome as occur in coccidiosis vaccination .
Please check spaces trough the manuscript : i.e line 309
Author Response
We would like to express our gratitude to Reviewer 2 for his insightful comments and thorough review of our manuscript. The authors completely agree with the comments and believe they contributed to the improved quality of our manuscript.
The following are our specific responses to the reviewer comments.
REVIEWER 1: I really appreciated your study for several factors. As first the originality: in general, the microflora of the respiratory tract of chicken is poorly investigated in spite of the numerous diseases affecting the chicken trachea, often with "collaboration" of a poor quality of litter. In this context, studies on turkey are more numerous. Another important factor is represented by the interrelationships among the gut, air and respiratory microflora influencing air and litter quality (also conditioned by possible treatments, as tumbling). I would define it as a real circuit. Your results are also supported by in-depth statistical analyses and discussed in exhaustive manner. Moreover, data relating to Escherichia coli and Gallibacterium spp. (Gallibacterium anatis) is currently considered an emerging pathogen) are interesting, also because of the potential role played as carriers/spreaders of resistance genes. Finally, although your results need to be further supported by other investigations, it seems clear that the environment air as well as the litter can influence microbiome of tracheal and intestinal tract respectively, although I believe that everything is strictly associated. Following you can find minor changes. I recommend you
- Material and methods section. It should be correct to specify vaccination plains. They could be an influencing factor for air and respiratory microbiome as occur in coccidiosis vaccination.
Response: Table S1 associated with the vaccination schedule was added and mentioned on Line 95 of the manuscript. Table S1 has also been included in the Supplementary material file.
- Please check spaces through the manuscript: i.e., line 309
Response: Line 23 and Line 287. The double spaces found were corrected (these were noted using the "Track Changes"):
Reviewer 3 Report
Just a few comments/recommendations. Overall, the manuscript is well written. Please check for a few English/grammar corrections. The experimental design and methodology are scientifically sound, the data are presented appropriately, and the conclusions are justified by the data presented. Litter management and the consequences of it are important to the poultry industry and the community at large. What goes on under different management regimes is well represented in the manuscript.
The reviewer does believe that the explanation of how many birds were used in the study should be made more clear - perhaps a table/flowchart.
Otherwise, no other comments.
Author Response
We appreciate Reviewer 3's feedback and have made the changes he requested.
REVIEWER 2: Just a few comments/recommendations. Overall, the manuscript is well written. Please check for a few English/grammar corrections. The experimental design and methodology are scientifically sound, the data are presented appropriately, and the conclusions are justified by the data presented. Litter management and the consequences of it are important to the poultry industry and the community at large. What goes on under different management regimes is well represented in the manuscript.
- The reviewer does believe that the explanation of how many birds were used in the study should be made more clear - perhaps with a table/flowchart.
Response: We added Supplementary Table 2 (Table S2) contains the information associated with the number of birds used and their distribution in the study groups and has been cited in Line 103. Table S2 has also been included in the Supplementary material file.
Therefore, we need to change the remaining supplementary table numbers: Lines 269, 272, 276, 279, 305, 314, 338, 352, 409, 498, 502, 589-594
In addition, the manuscript has been reviewed by a native speaker of the English language, Dr. Helen Lowry; she has been incorporated in the acknowledgments of our manuscript, Lines 614 and 615.
The superscript number of the CFU number was incorporated into the Lines 81, 99, 238, 239.
English/grammar corrections were added in Lines 39, 308, 310, 334, 409, 490, 499, 517, 581-584.